# Give Older Persons a Voice in the Society—By Using Information Compiled during Preventive Home Visits on a Societal Level

**DOI:** 10.3390/ijerph18147433

**Published:** 2021-07-12

**Authors:** Anna Nivestam, Maria Haak, Albert Westergren, Pia Petersson

**Affiliations:** 1The Research Platform for Collaboration for Health, Faculty of Health Sciences, Kristianstad University, SE-291 88 Kristianstad, Sweden; maria.haak@hkr.se (M.H.); albert.westergren@hkr.se (A.W.); pia.petersson@hkr.se (P.P.); 2Department of Health Sciences, Faculty of Medicine, Lund University, SE-221 00 Lund, Sweden

**Keywords:** aged, communication, decision makers, determinants of health, health promotion, inclusive society, monitoring, older people, old age policy, participation in society

## Abstract

Preventive home visits (PHVs) are offered to older persons with the purpose of promoting health and preventing risks on an individual level. However, aspects of health need to be considered on a societal level as well. This study aims to get a deeper understanding of perceptions of the usability of the information compiled during the PHVs to promote health, among older persons, on a societal level. Three online focus groups were conducted with heads of unit of PHVs, heads of department, and politicians responsible for health and welfare in seven municipalities in Sweden (*n* = 12). The findings were visualised in the core category *Enable an inclusive society* and the interrelated categories *Monitoring determinants of health* and *Enabling exchange of information*. The information from the PHVs could be used to monitor determinants of health by identifying assets, challenges, shifts, trends, and future needs in the society. Moreover, exchange of information from the PHVs could occur within and outside the health and welfare organisation. However, the potential use was affected by hindrances illustrated in the category *Obstacles to interpreting and communicating the information*. To conclude, using the information from the PHVs could possibly contribute to an inclusive society, where persons not usually represented in decision making are given a voice.

## 1. Introduction

Preventive home visits (PHVs) are offered to older persons with the purpose of promoting health and preventing risks on an individual level. However, to promote health, aspects of health need to be considered on a societal level as well. According to a model used for PHVs in Sweden, information about, for example, older persons’ lifestyles is gathered and aggregated by the municipalities. Hence, more knowledge is needed to take advantage of the information gathered during the visits and to create a society which helps older persons to maintain or improve their health. 

The number of persons at an old age is increasing all over the world [1], and in 30 years, the number of persons 65 years or older will double [2]. In addition to this, the COVID-19 pandemic has led to increased social distance and isolation among older persons [3,4]. This situation requires those in leading positions to create a society which promotes health among older persons. To promote health in old age, an age-friendly world is suggested by the World Health Organization (WHO) [5], which emphasises the need to create an environment that is accessible, offering housing which is affordable and enabling lifelong learning, to give a few examples of determinants of health. In 1991, Dahlgren and Whitehead described determinants of health that have an impact on a person’s health status. They described the environmental determinants displayed in mainly three layers around the individual determinants (e.g., age and sex). The first layer is material and social conditions, such as education and housing. The next layer is support from the social networks, and the final layer is individual lifestyle factors [6]. Later, in the health development model, Bauer et al. [7] described the interaction between the individual and the society, where the person’s health status affects their ability to have an impact on the societal environment and vice versa [7]. Moreover, to promote health, the health development model emphasises a multilevel approach involving both the individual and the society with its environment around the person. However, even though both an individual and societal perspective are needed, health interventions tend to be focused on the individual level and the person’s behaviours to improve or maintain health and lack a multilevel approach [8,9]. 

In general, PHVs are offered around the world with the purpose of promoting health and preventing risks, with a focus on the individual level [10]. One health intervention established in the south part of Sweden, offers PHVs to older persons (77 years or older without home care) with an opportunity to promote health and prevent risks both on an individual and a societal level [11,12,13]. Previous research has focused on the effects of PHVs, for example, on mortality [14,15,16,17], use of health and social services [15,16,18] and different individual health outcomes [15,16,17]. Furthermore, research has looked into older persons’ [19,20] and health professionals’ [21] experience of the visits. Advantages of PHVs, such as reduced mortality and functional decline, have been shown in a systematic review [10]. Moreover, a recent study showed reduction in societal costs after PHVs [22]. However, to our knowledge, research has not considered how PHVs could help those in leading positions to promote health from a societal perspective. Nivestam et al. [13] highlighted in a study that older persons felt included in the society due to the support given during the PHVs. Moreover, another study showed that factors such as the ability to do things that make one feel valuable and the ability to participate in social activities were associated with good health among older persons who received a PHV [11]. However, it remains unclear how such factors identified during PHVs could be considered on a societal level to promote health. Thus, further research is needed to investigate how society could take advantage of the information gathered during the PHVs to create a society which helps older persons to maintain or improve their health. The present study focused on those in leading positions within the municipality. The aim was to get a deeper understanding of their perception of the usability of the information compiled during the PHVs to promote health, among older persons, on a societal level. The study aimed to answer the following two questions: (1)How can information be used to promote health among older persons on a societal level?(2)Are there any obstacles to the potential use, and if so, what are they?

## 2. Materials and Methods

The focus group method was used in this study, where the purpose is to create a collective understanding of a phenomenon [23]. In focus groups, knowledge is created out of the discussions built on interactions between the participants. This method suited the present study’s aim well, because promoting health on a societal level is a collective responsibility and cannot be dealt with by single individuals. Therefore, we were interested in how the focus groups perceived the usability of the information compiled during the PHVs. To highlight the participants’ discussions, the researcher encouraged the participants to talk to each other and, during the discussions, asked probing questions to help them further elaborate on the subject. Moreover, to create a talkative atmosphere during the focus groups where the participants feel comfortable and share a common frame of reference, it is suggested to create homogeneous groups [24]. Therefore, persons from the same leading position were recruited into the same focus group. In addition, to get a variety of perceptions a heterogenic sample is recommended [24]. To achieve that, persons from different leading positions with different responsibilities within the municipalities were invited to take part. 

### 2.1. Context

The model for PHV used in the south part of Sweden for five years has been jointly developed within a research and collaboration project between seven municipalities, the county council, and Kristianstad University. A steering committee, which consists of the heads of unit for the PHV in each municipality and representatives from the county council and Kristianstad University, meets four times every year to consider the project’s progress. Furthermore, those who conduct the visits and older persons have been involved in development of the questions asked during the visits. According to the jointly agreed model, PHVs are offered to older persons (77 years or older without home care) in the seven municipalities. Visitors who are health professionals, such as district nurses or assistance nurses, conduct the visits. During the visit, which lasts for approximately two hours, a dialogue is created guided by predetermined questions, about, for example, nutrition, physical factors, mental factors, housing, and finances. The questionnaire includes mostly closed and a few open questions, the purpose being to focus on the dialogue and reduce the amount of time spent on writing. Answers to the questions are digitally registered by the visitor during the PHV. These data are compiled by the municipality and can form the basis for strategic decisions. Examples of information from the PHVs are descriptive statistical data, single opinions or thoughts of older persons which evolve from the dialogue, and the visitor’s observations or assessments during the visits. 

The society includes a diverse range of persons in leading positions, at local, regional, and national levels [25]. This study takes its starting point in persons in leading positions at local level, i.e., politicians and employees who have the mandate to take decisions within the municipalities’ health and welfare organisations and have knowledge of the PHVs. In the seven municipalities, the health and welfare organisation is responsible for providing the PHVs. The health and welfare organisation consists of a committee with politicians and a department with employees from the municipality. The municipalities in Sweden are self-governed according to the Swedish Local Government Act [26] by politicians elected every fourth year. This means that the health and welfare department is governed by politicians in the committee that makes decisions about the PHVs. Employed leaders within the municipalities’ health and welfare department are heads of unit for PHVs and heads of departments.

### 2.2. Sample and Procedure

Purposive sampling was used to recruit participants (Figure 1). The sample was homogeneous in terms of representing those in leading positions within the municipalities that offer PHVs. In terms of variation and to capture diversity in perceptions, persons within leading positions from three different levels (heads of unit for PHVs, the chair of the political health and welfare committee, and heads of department responsible for health and welfare) were recruited.

Participants were identified through the municipalities’ official websites (politicians, heads of department) and through the steering committee for the PHVs (heads of unit). Three months before the focus groups, politicians (in total *n* = 9) and heads of department (in total *n* = 9) were contacted by email. In two municipalities, the health and welfare organisation was divided into two organisations. Therefore, nine politicians and nine heads of department were invited to take part. Those who did not respond to the email were contacted by phone within two weeks from the first email. To those who did not answer after the first email or repeated phone calls, a second email was sent out after three weeks. The third group with persons who were heads of unit for PHVs (*n* = 7) in each municipality were recruited and informed about the study at a regular steering committee meeting during the autumn and by email for those who did not attend the meeting. Persons who declined to take part or did not respond to the contact were considered as dropouts. Approximately one month before the focus group session, all who agreed to participate received a digital calendar invitation with the following: a web-link to enter the online meeting, an informed consent form, background questions, and an agenda showing the purpose of the study and the areas to be discussed. First, three separate focus groups (Figure 1) were conducted, one with heads of department (*n* = 4), another with politicians (*n* = 3), and a third with heads of unit for PHVs (*n* = 5) (Table 1). Lastly, the preliminary findings were presented and discussed at a seminar with representatives from all three groups (*n* = 6) as a member check [27], and then at a scientific seminar with researchers experienced in qualitative research.

### 2.3. Development of the Questioning Route

In accordance with Krueger and Casey [24], a questioning route was developed. First, questions related to the aim were generated among the authors based on previous research and experience from working with the jointly model for PHVs. Second, open, transition, key, and ending questions were identified. Third, the questions were formulated in an understandable way. Krueger and Casey [24] recommend testing the questions on coworkers in order to avoid taking extra time from the participants. Therefore, a tentative focus group session was conducted with coworkers (*n* = 5) lasting for one hour. After the session, coworkers were asked for feedback about the questions and procedure. To improve the procedure, in accordance with the feedback, a vignette was developed containing data from the PHVs. Moreover, the tentative focus group was recorded, and afterward, the authors discussed the questions and the procedure in order to become thoroughly comfortable with the procedure. The final questioning route was assigned the following discussion areas: information content and need, utilisation of information, information transfer, benefits, and drawbacks of PHVs.

### 2.4. Data Collection

Three focus groups were conducted between January and February 2021, via an online video communication platform. This communication platform was familiar to the participants as they used it in their daily work. First, one focus group with heads of department was conducted, then one with heads of unit, and lastly, one with politicians. The focus groups were moderated by the last author and comoderated by the first author. The last author introduced and guided the focus group and asked probing questions for clarification. The first author observed the interaction between the participants, asked probing questions, and took notes during the focus groups. After each session, the last author and the first author had a debriefing about their experiences and wrote reflective notes. The focus group sessions lasted for 1 h 40 min, 1 h 30 min, and 1 h 10 min, respectively. In advance, the participants were asked to fill in a form requesting background information (age, gender, education, and year in the current position) and informed consent. Before the focus group started, written and oral informed consents were received from all participants.

All three focus groups were conducted in the same way. First, the moderator welcomed the participants, introduced the purpose of the study, and asked the participants to ensure their microphone and video were turned on during the whole session to facilitate the discussions. Thereafter, the session began with a presentation of the participants, and the researchers explained their role. The moderator emphasised that the participants were the experts, and their perceptions and discussion were of interest. The moderator’s role was to facilitate the discussion. As a starting point for the dialogue, the first author presented a five-minute vignette about the model for PHV, described previous research about PHVs, and presented some data gathered during the PHVs. The data presented concerned, for example, persons’ use of smartphones and computers, persons’ taking part in associations and activities, and loneliness. All focus groups were subject to audio and visual recording and afterwards transcribed verbatim by a professional transcriber.

### 2.5. Data Analysis

Data were analysed according to the approach described by Krueger [28,29]. The main purpose of the method is to capture the essence of the discussions rather than individual aspects. Emphasis was on analysing the data with the discussions in focus. The aim of the study and the research questions guided the analysis. The notes taken during and after each focus group were used to support the interpretation process. NVivo 2020 software was used to organise and sort the transcribed material into discussions and later into categories. Krueger [28,29] suggested doing the analysis according to the following steps. First, the transcripts and audio-recorded material from the focus groups were read and listened to several times to get an overview of the data, and potential trends and patterns were observed. Thereafter, the first author and the last author separately identified content in the discussions that related to the research questions of the study and provided a brief descriptive summary of the content. Next, the researchers compared the identified discussions with each other’s to verify the findings. Thereafter, the first author and the last author interactively identified and synthesised the descriptive meanings of the discussions into subcategories and categories, which provided a more abstract level of the findings than the summary of each discussion content. The research question ‘Are there any obstacles to the potential use, if so what are the obstacles?’ resulted in one category with subcategories. Thereafter, the categories which answered the research question ‘How can information be used to promote health among older persons on a societal level?’ were combined and interpreted to get a deeper understanding of the findings, represented in the core category. The process was thereafter reviewed and verified by the co-authors. As a member check, all participants were invited to a seminar led by the first author and the second author where the preliminary findings were presented and discussed. The seminar confirmed the findings, and no new information emerged. Thereafter, relevant quotes were identified, which reflected discussions and interactions from all focus groups. Lastly, a validation of the findings was carried out during a scientific seminar that contributed clarification and improved the vocabulary. The ultimate findings were examined by all the authors and agreed upon. The consolidated criteria for reporting qualitative research (COREQ) were used to report the study [30].

### 2.6. Ethical Considerations

Written and oral informed consents were obtained from all participants and the study was conducted in accordance with the Declaration of Helsinki [31]. According to Swedish legislation [32], there is no need for ethical permission for this type of study, since no sensitive personal data are collected, and there was no obvious risk for any person taking part in the study. It could be an issue to assure confidentiality in focus groups, due to that the researchers cannot guarantee that the participants do not pass on what was discussed [33]. Additionally, there might be an increased risk of inhibiting the confidentiality by using an online communication platform. However, in this study, the subject for the discussions was not of a sensitive nature, which diminish the risk of harm. Moreover, there is a risk of third-party harm when discussing subjects related to other persons (older persons in this case) not present in the discussion. To consider the risk of third-party harm, discussions were kept on a public level, and in case anyone brought up issues related to specific persons, the moderator was alert to change the direction of the discussion.

## 3. Findings

The findings develop the understanding of how the participants perceived the potential use of the information from the PHVs and what obstacles affected the potential use. How participants perceived the use was visualised through the core category *Enabling an inclusive society* (Figure 2). The participants agreed that information from the PHVs could to a greater extent be used to give older persons a voice. The statistical data represent older persons subjective answers to the questions asked during the PHVs, and by using the statistics, older persons could indirectly be given a collective voice. In addition, older persons’ individual thoughts expressed during the PHVs could give them a direct voice. This information could then be utilised on a societal level by persons in leading positions to discover health determinants and take action to promote health. Interrelated with the core category were two categories exemplified in five subcategories. The category *Monitoring determinants of health* represents the use of information, primarily in terms of statistical data, for identifying assets, challenges, shifts, trends, and future needs, in the society. The category *Enabling exchange of information* represents the use of information from the PHVs for internal and external communication within and outside the own organisation, and the participation of older persons. However, the potential use was affected by hindrances illustrated in the category *Obstacles to interpreting and communicating the information*, which influences the monitoring of determinants of health and the information exchange. This third category is visualised in two subcategories, which reflect a need for improved understanding of the descriptive statistical data and highlight that the information tended to be retained by the health and welfare organisation. 

### 3.1. Monitoring Determinants of Health

This category shed light on how information in terms of statistical data from the PHVs could be used for monitoring determinants of health. The discussions focused on the health determinants: material and social conditions, social networks, and lifestyle factors. From the health determinants the participants identified assets, challenges, future needs, shifts, and trends in the society with help of the PHVs data, as illustrated in two subcategories.

#### 3.1.1. Identifying Societal Assets and Challenges

The participants reflected upon how information via the PHVs could be used to identify societal assets and challenges. From the perspective of the participants, health determinants such as digital technology use, engagement in associations, public transports use, and housing availability could be identified as societal assets or challenges with the help of the PHVs data. Thus, the data gave the participants insights about actions that needed to be taken to include or reach out to older persons. The participants agreed that the PHVs data could help them visualise a low level of engagement in associations, and they reflected on, for example, how to reach those not taking part in societal activities today. Lack of digital-technology use among a proportion of the present population, was identified as a challenge that needed to be considered by the municipality to avoid social exclusion. The quote below illustrates a conversation between heads of department about potential challenges which could be identified with the help of the information:


*(Head of department X) We can use it as a basis for what we should invest in, in the municipalities/…/, but you then see that 43 percent do not have a smartphone or do not use a computer [data from the vignette]. It is true in several municipalities today that you can learn; you can go somewhere, and learn about your phone and your computer/…/*



*(Head of department Y) Just smartphone courses and learning how to use your tablet and stuff like that; it has actually been the pensioners’ associations in our municipality that have done quite a lot on this, but then you get to the point that it is no more than 48 percent… who are part of an association. And if you then do not have a computer or smartphone or are not part of PRO [a pensioner association], it does not help that the association does a lot.*



*(Head of department Z)/…/we have also seen this in the preventive home visits, that there are quite a few who are still not so comfortable with digital technology.*


#### 3.1.2. Identifying Shifts, Trends, and Future Needs in the Society

The participants’ perceptions about the use of information as an aid to identifying shifts, trends, and future needs in the society, regarding lifestyle factors and housing were highlighted in this subcategory. According to the participants, data from different years could be compared and used to visualise how determinants of health develop over time. The participants believed that data from the PHVs could show discrepancies in health determinants and whether a determinant tended to increase or decrease. Furthermore, the participants claimed that with help of the data, they could detect generational shifts in health determinants, for example, when it came to lifestyle factors. They expected that the generational preferences could change, and if this were seen in the data, they could take actions within the municipality. Moreover, the participants agreed that identifying shifts in a health determinant related to an action could be an evaluation and could confirm if the action had any effect or not. In addition, they discussed whether data could be used to give guidance about future health determinants. As an example, they discussed how, with the help of the data, they could identify the need for future housing, such as senior apartments and retirement homes and could visualise older persons’ wishes regarding housing opportunities. The data could help participants identify the present housing availability, which might need to be considered to cover the future requirements for accessibility and types of housing features (elevators, number of rooms, and in which area they want to live) this population has and values. Agreement between heads of unit for PHVs about the usefulness of the information to identify future needs was expressed as:


*(Head of unit X) We have a concern about the fact that we have an ageing population and many live in their houses and the houses are not very accessible, but we also do not have housing that can meet [the older persons] need and we need to know what they [older persons] think; i.e., where do they want to move to and how do they want to live, if they can no longer live in their house? So there are a lot of things that we think about and I know that they asked some of these questions during the home visit today/…/*



*(Head of unit Y) Yes, thus this is a form of planning… thus, to use it as a planning tool or as…as a basis for decisions.*


### 3.2. Enabling Exchange of Information

The focus groups developed an understanding of how information from the PHVs could enable the exchange of information in different ways: internally within their own organisation, and externally outside the organisation and with older persons, meaning that information from the PHVs could be exchanged through communication among employees in the municipalities, politicians, stakeholders, and older persons. The focus groups revealed that employees, politicians, and older persons were dependent on each other and needed to discuss the information from the PHVs to generate actions. This category, *Enabling exchange of information*, is illustrated in three subcategories.

#### 3.2.1. Enabling Internal Communication 

This subcategory represents the participants’ discussions about the use of information to enable internal communication within the health and welfare organisation. The participants perceived that information could enable communication between visitors conducting the PHVs, heads of unit, heads of department, and politicians within the organisation. On the political level, participants believed that statistical data from the PHVs could be presented by the visitors, heads of unit, or heads of department and thereby enable communication between politicians and employees within the organisation. In addition, the participants talked about the need for a presentation made by the visitor about the observations or assessments made during the visits but not necessarily documented in the digital system. On the operative level, within the organisation, the participants highlighted that data could constantly stimulate communication between the visitors and the heads of unit. The information acquired by the visitor could be brought up in a dialogue with the heads of unit and thereby bring about change within the organisation; for example, with the help of this information, they could modify activities such as group size in social activities and how information about the municipality could reach older persons. This quote exemplifies a discussion in the political focus group:


*(Politician X) … the employees they have perpetually in their assignment, are there things that we see are not good, so to speak…do not get better or something like that? Then they have to work on it and they do. So that we do not lag behind in anything, the purpose is always that things should be better than they are today.*



*(Politician Y) The important thing is that the profession helps the politicians with the analysis…*



*(Politician X) Yes it is.*



*(Politician Y) … that they present it to us and also tell what…to some extent what it means. Then we will of course look and try to make our own opinion as far as possible.*



*(Politician Z) Yes, it is the person who works on these issues [the visitor], as I mentioned the visitor earlier who have worked for many years with this [the PHVs], has to come to the committee more often/…/and [we can] listen to the visitor and hear the visitor’s thoughts, what are we missing, what we need to work on. So I would like to say that, maybe there is no need for surveys, but more a need to listen to the visitor with the experience he or she has.*


#### 3.2.2. Enabling External Communication 

The participants emphasised that the information from the PHVs could enable external communication outside their own organisation. To give a few examples, the participants agreed that the external communication could occur with employees, stakeholders, and politicians outside the health and welfare organisation. The participants suggested using the information from the PHVs in communication with the county council and among municipalities, which in turn could generate common solutions to shared challenges. This subcategory, *Enabling external communication*, highlights that PHV data could be used in communication with other political committees and as a starting point for communication with other organisations in the municipality. For example, the participants suggested communicating the information to persons responsible for public health, infrastructure, planning, and building. Moreover, the participants perceived that data from the PHVs could enable discussions with external partners, such as building contractors. The following quote shows a conversation between politicians about how the external communication could occur: 


*(Politician X) Yes, I think if there is someone who experiences obstacles to getting out and being more involved in society, there might be things that we can re-construct or facilitate in some way. To make it easier to get to meeting places or whatever it may be that is the concern, so to speak.*



*(Politician Y) We had a dialogue with them [building company], so now they are building 28 new senior houses in the city-centre, just for older persons… it [the senior houses] ends up in the middle of the city-centre, where we have activities, and it [from the houses] is about 50 metres to the largest grocery store and close to the bus station and so on…*


#### 3.2.3. Enabling Participation of Older Persons 

Information from the PHVs could enable the participation of older persons, and two ideas emerged concerning how this could be done. First, the participants discussed whether the information from the PHVs could per se represent voices from older persons, because during the visits, older persons could obtain information and guidance on how to communicate with those in leading positions, for example, through citizens’ proposals. Second, the participants highlighted that the PHV data could be used in conversations with older persons, and thereby, solutions to the challenges shown in the data could be found. The participants said that the information generated through the PHVs added a broader dimension to the existing voices represented in pension organisations. They emphasised the value of listening to older persons’ voices and thought that the information from the PHVs revealed a dimension which could not be captured in other surveys. They also considered that the information reflected voices from persons not usually involved in decision making. On the whole, the participants expressed the view that the exchange of information between older persons and those in leading positions could add a democratic value to the visit, as shown in the following quote: 


*(Head of department X) [at the home visit]…it turns out that you have an opinion…then the visitor can take that information [to the health and welfare organisation] or you send it [to the organisation], or the visitor show…how do you give your opinion to the municipality [digitally], because then they [older persons] will be able to leave other opinions later on as well… Otherwise we [the health and welfare organisation] will receive them orally or written by hand and then we will do something with them. So I think that…this is probably the best way, then you also get feedback from the person who…who is to handle the matter.*



*(Moderator) How is it in the other municipalities? Do you also have such [system for] opinions…?*



*(Head of department Y) Yes, absolutely, and I think that we…and it is the same then as you [Head of department X] say.*



*(Head of department X) Is there any general point of view or so that you want to put forward, it is done the same way, that the visitors help them leave that point of view to the right person.*



*(Head of department X)] It [the opinions] is high and low and it should be high and low, that is how democracy works and…*



*(Head of department Y) Yes.*



*(Head of department X) … transparency also, that you can ask what questions you want and have opinions on whatever you want.*


### 3.3. Obstacles to Interpreting and Communicating the Information

The participants perceived obstacles to interpreting and communicating the information, which is visualised in this category. In the focus group discussions, they called for an improved understanding of the statistical data and highlighted that the information tended to be retained by the health and welfare organisation. From the participants’ perspective, statistical data from the PHVs did not give them the full picture, and they stressed the need for deeper analyses and qualitative information to better understand the statistics. In addition, different obstacles for communicating the information outside their own organisation were highlighted.

#### 3.3.1. Calling for Improved Understanding of the Statistical Data

The participants highlighted obstacles to interpreting the information, and they called for an improved understanding of the descriptive statistical data. They said that to make full use of the data, deeper statistical analyses were needed to investigate causal effects, but also qualitative information was required to capture the essence of a complex phenomenon. They reflected on how difficult it was to capture loneliness with one single question. Therefore, the participants suggested that to understand what could be done on a societal level, more qualitative information was needed. In addition to more qualitative information, deeper statistical analyses were needed to be able to fully monitor determinants of health. Moreover, during the focus groups, they discussed the difficulty of having time and the right prerequisites to analyse data within the municipality. The participants suggested asking researchers for help with deeper analysis of the data, for example, by putting together qualitative information acquired by the visitors in a systematic way. In the following quote, heads of department talked about the obstacles to the interpretation of the statistical data: 


*(Head of department X) … I think of these, 21 percent who experience physical problems that prevent them from participating. Is it because they think it is difficult or is it because there is poor accessibility or…they do not want…so this [the statistics] asks a few follow-up questions.*



*(Head of department Y) Yes, so in the analysis it can be [good to get a more in-depth analysis], because otherwise I think we can add things that we think are reasons that they do not participate.*


#### 3.3.2. Being Retained by the Organisation

Obstacles to the use of the information were discussed as information tended to be retained by the organisation, meaning that it could be difficult to communicate the information outside the health and welfare organisation. According to the participants there could be different reasons for why it was difficult to communicate the information gathered during the PHVs outside the organisation. The reasons suggested were lack of interest from other organisations outside the health and welfare organisation in talking about issues related to older persons’ health or due to norms which implicitly express that older persons’ health was a responsibility for the health and welfare organisation. At the same time, the participants felt that older persons’ health was something for the whole society to consider. Other suggestions mentioned were the overload of other types of data and that other organisations might not see the value of more data. Moreover, the participants speculated that it could be because different organisations use other types of language or valued other things, or the physical distance between the organisations. However, the participants did not come to a clear explanation of why it was hard to communicate the information with others outside the organisation. On the other hand, the participants considered different suggestions to facilitate the access to and use of the information generated from PHVs, such as creating a communication plan for when, how, and to whom communication of information should occur. In the following quote, heads of unit for PHVs discuss the obstacle of information being retained by the organisation: 


*(Head of unit X) …the results that come in from the preventive home visits, just this with how the older persons want it [to live] …we use it [the information] in our thinking in my organisation [health and welfare], for example [information about] involuntary loneliness, which is made visible [through the preventive home visits]. Of course we can use it [the information] to get tips and ideas, how do they want it [e.g., type of support, activities], to be encouraged to leave their homes to come to our activities. But it is in our small organisation that we can use it, but there is a different perspective and that is the bigger one, I am thinking about [how information can be used in] societal planning.*



*(Moderator) Yes.*



*(Head of unit X) But we have not achieved that…*



*(Head of unit Z) Yes, I was just going to agree with it [what the head of unit X said], it is easy to use it [the information] within your own organisation, but it is hard to find ways and use this information [from the PHVs] in the other parts of the municipality… but this is about health and care, it concerns the older persons. But so do societal issues, they concern the older persons, but it…there is a strange obstacle there. It would be interesting to know what it is that really does it [makes it difficult to transmit the information outside the organisation].*


## 4. Discussion

The present study focused on the usefulness of the information compiled during the PHVs as perceived by persons in leading positions. Our main finding showed that by monitoring determinants of health and enabling information exchange, for example, between the municipality and older persons, information from PHVs could be used to enable an inclusive society. However, using the information from the PHVs could be hindered by obstacles that influenced the interpretation and communication of the information.

The present study showed that the information gathered during the PHVs could enable an inclusive society. Older persons could be given a voice, indirectly through the statistical data and directly through listening to older persons’ thoughts expressed during the PHVs. In accordance with the health development model, a person’s health status can influence the ability to take part and have an impact on the society [7]. Previous research highlighted that a diverse range of older persons do not participate in societal practice and decisions due to barriers such as mobility issues, hearing difficulties, and lack of self-confidence [34]. Moreover, another study showed that older persons who took part in decision making within the society were usually healthy and did not represent a diverse older population [35]. However, the present study showed that by using the information from the PHVs, persons not represented in decision making today could be included, despite health challenges. The focus group discussions emphasised that by using the information from the PHVs, a more diverse group of persons could be included in decision making, a group that is not represented in the organisations that represent older persons’ opinions today. Nevertheless, one must be aware that the information from the PHVs does not represent all older persons in the context of this study. For example, persons who have home care are not included in the population who are offered PHVs. Thus, to be able to progressively develop towards an inclusive society for all older persons, voices from a more diverse population must be considered. By using the information from PHVs, one step towards an inclusive society is possible.

However, older persons’ level of inclusiveness could be questioned. The United Nations has stressed the importance of an inclusive society in its sustainable development goals, especially goal number 11 (make cities inclusive, safe, resilient, and sustainable) [36]. However, a study that analysed international policy documents showed that older persons’ participation in societal decisions was often stressed as important in policies [37], though the level of participation could be questioned. The study showed that participation was expressed in the policies as communicating with older persons, though, how this should occur was unclear [37]. As well as in policies, the focus group discussions in the present study identified communication with older persons as important, particularly having a dialogue about the information compiled during the PHVs and listening to their proposals. However, according to Arnstein [38], consultation can be seen as tokenism (someone is listening but does not give the person talking any actual power) and has no actual impact on decisions taken. Nevertheless, it is not feasible to include all persons in the decision-making process; therefore, the participation of older persons suggested in the present study might be an alternative way to increase inclusiveness. However, to make sure that older persons are listened to and given the power to have an impact on decisions, a more explicit communication strategy has to be developed.

To enable an inclusive society, determinants of health must be discussed outside the health and welfare organisation. The present study visualised a possibility to use the information from PHVs in external communications, for example, with persons responsible for public health, infrastructure, planning, and building. However, at the same time, obstacles to the external communication that were highlighted as information tended to be retained by the organisation. The present study showed that one reason for this challenge in communication could be norms, meaning that older persons’ health was viewed as a responsibility solely for the health and welfare organisation. Even though it has been highlighted for decades that health is an issue for all sectors of society to consider, still, considering health in all sectors within the society seems challenging in practice. As early as 1986, the Ottawa Charter highlighted that promoting health must be considered by the whole society [39], and since then, it has been repeatedly highlighted in international policies [36,40]. Another obstacle expressed in the present study that could hinder the external communication was that the terminology used might not be understandable in all sectors within the municipality. This was also recognised in a study from Norway, which found that the terminology used in health promotion was difficult to grasp for sectors not directly working with health [41]. In addition, a review by Corbin et al. [42] highlighted multiple aspects that facilitated intersectoral work with health promotion, for example good leadership, shared goals, communication, and contextual awareness. With help of this research [42], the obstacles to communicate the information outside the own organisation might be solved. However, none of the studies in the review by Corbin et al. [42] addressed health promotion in old age. Thus, more research is needed to investigate how external communication could be facilitated. Facilitating external communication could later lead to collaboration that promotes older persons’ health.

The focus group discussions highlighted a need for qualitative information from the PHVs, for example, the visitors’ observations during the visits and older persons thoughts expressed during the PHVs. Qualitative information seems to be as important as statistical data to monitor determinants of health and generate actions that promote health among older persons on a societal level. However, a recent review showed that data for monitoring health in the society is usually quantitative [43], which might facilitate the identification of shifts and trends. The present study highlighted a need for qualitative information to get a deeper understanding of the statistics and thus fully monitor health and understand older persons’ needs. There seems to be a lot of valuable information which cannot be captured by statistics and the statistics raise new questions about what actions should be taken to deal with challenges. The focus group discussions revealed that the solution to this was to enable participation of older persons, meaning for example inviting older persons to discuss the challenges visualised with help of the statistics. Hence, to utilise the qualitative information, there is a need for more knowledge about how to systematically compile the qualitative information.

To sum up, it is time to turn thoughts into action. The present study visualises different possibilities for how information from the PHVs could be used to enable an inclusive society. However, to make sustainable change and to put the ideas into action, help can be gained from research about knowledge translation. In previous research, the concept of knowledge translation has been used to clarify how to use research in practice, and different frameworks that could help this process have been displayed, for example, the WHO ageing and health knowledge translation framework [44]. Seven elements (e.g., context, relationships, push and pull efforts, and evaluation) are highlighted in the ageing and health knowledge translation framework, which can be reflected upon to facilitate the use of information [44]. This framework highlights for example a need for an assessment of the context where in this case the information from PHVs could be used. By doing this, the possibility to relate the information from PHVs to other sources of information used for decision making in the municipalities increases. Further, before taking decisions that affect older persons, it is important to consider the cultural context to enable inclusiveness, for example, how older persons are viewed and who is getting heard and listened to. This contextual analysis could situate the information in the existing municipality’s context. Persons in leading positions can possibly be inspired by the framework for knowledge translation in their continued work to take action in using the information compiled during the PHVs to promote health. 

To establish trustworthiness in the present study, methodological considerations were discussed, as inspired by Lincoln and Guba [27]. The focus group method was chosen to increase the possibility of achieving the objective of the study. Through the focus group sessions, the participants came up together with new ideas for potential use of the information, ideas which would not be captured in individual interviews. Furthermore, a purposive sampling method was used to reach persons who could best discuss the phenomenon under study and give a variety of perceptions. The participants were a mixed group in terms of gender and time in current position; they came from different municipalities, and there were three levels of leaders. However, the sample was quite homogeneous in regard to age; a younger age group might have had other perceptions not captured in the present findings. To ensure a permissive, nonthreatening atmosphere, which generated fruitful discussions, the participants were assigned to different groups based on leading position. The number of participants in each focus group was quite low due to some dropouts. However, the discussions were very rich, and the number of participants seemed to be sufficient to stimulate discussions online. 

The vignette which was used to introduce the focus group discussions may have influenced the subjects of discussion. However, the vignette was an appetiser and motivated the participants to start the discussions. Moving on, let us discuss the online communication platform’s impact on the focus group session. Due to the COVID-19 pandemic, the use of online communications platforms has increased [45]. Persons involved in this study were using this communication platform for meetings in their job, which facilitated the procedure of this study. Another strength of using an online communication tool is that it saves time on travelling, which might increase the chances for the participants to take part. A challenge of online communication instead of live meetings is the difficulty in making eye contact and seeing the full body language, and this could have had an impact on the communication between the participants. 

To further increase the trustworthiness, discussion contents related to the aim were identified by two authors separately, and the findings were verified by all co-authors. Moreover, quotes were chosen to verify the findings. A member check was performed, which represented persons from all three focus groups. The preliminary findings were presented and discussed during this member check, which confirmed that the findings were true for the participants. This final discussion increased the possibility for the participants to add anything that they missed out from previous focus group sessions, but no new information came up during the member check.

The findings may be contextually bound since this model for PHVs might not be the same somewhere else. However, the findings can give an indication of how information from other similar sources can be used on a societal level. In addition, the present study highlights obstacles to the use of the information that might exist in other situations. Given the carefully constructed sampling procedure with a variation of participants, the findings might be transferable to similar contexts where societal decisions are taken to promote health among older persons. Persons in other leading positions in the municipality were not involved in this study, for example, persons outside the health and welfare organisation. Including persons from other leading positions could capture other aspects of the use not seen in the present findings.

## 5. Conclusions

This study adds a novel perspective of the PHV which has never been studied before. Information gathered during the PHVs seems to be valuable for those in leading positions to use in order to create a society which promotes health. By using the information on a societal level, PHVs become a multilevel intervention that can promote older persons’ health both on an individual and a societal level. This study highlights new ways for potential use of the information (in terms of statistical data, visitors’ observations and older persons’ thoughts) generated during the PHVs, to promote health on a societal level. By using the information, those in leading positions within the municipality could enable an inclusive society that represents older persons who do not usually take part in decision making. The information could be used for monitoring determinants of health and for information exchange. However, obstacles exist to interpreting the information and communicating it outside the health and welfare organisation. These obstacles need to be considered in the future work on the use of the information compiled during the PHVs.

## Figures and Tables

**Figure 1 ijerph-18-07433-f001:**
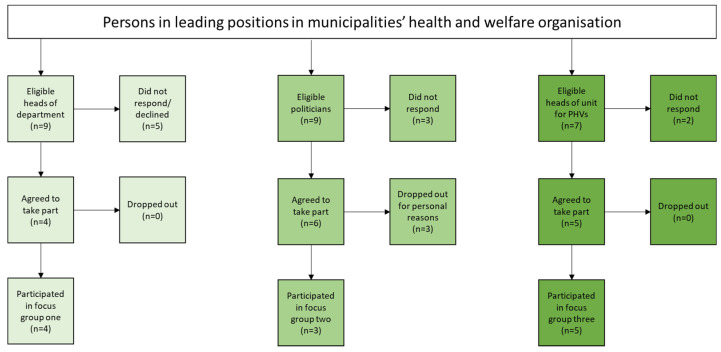
Overview of the sampling procedure. Three separate focus groups with heads of department for health and welfare, politicians responsible for the political health and welfare committee, and heads of unit for the preventive home visits (PHVs) were conducted.

**Figure 2 ijerph-18-07433-f002:**
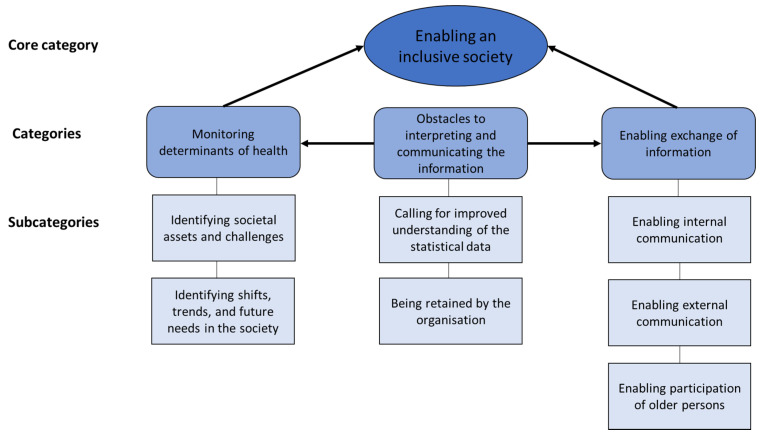
The core category *Enabling an inclusive society*, with its categories, and the category *Obstacles to interpreting and communicating the information*, which influences the other categories.

**Table 1 ijerph-18-07433-t001:** Characteristics of the sample.

Characteristics	*n* = 12
Men/women, *n*	5/7
Age, median (rang) ^a^	51 (44–60)
Highest education, *n*	
≥3 years university	6
University <3 years	4
Upper secondary school	2
Years in current leading position, median (rang)	5 (1–25)

^a^ Three missing values on age.

## Data Availability

To preserve the participants’ privacy, the data are not publicly available.

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
