# Peer review of "Give Older Persons a Voice in the Society—By Using Information Compiled during Preventive Home Visits on a Societal Level"

_ijerph, 2021, doi:10.3390/ijerph18147433_

Round 1

Reviewer 1 Report

This work proposes ways to improve the welfare of the elderly in Sweden, especially health, using PHV. This study is not a technical study, nor does it propose technical specific information. However, this work is a valuable and comprehensive study that suggests how PHV, an information gathering strategy, can be used for the welfare of the elderly.

This study is a comprehensive study, and there are limitations that cannot be used in tables or pictures and are forced to be expressed in long sentences. However, to improve the overall completeness of the paper, I recommend revising the following.

-Describe in detail the justification or advantages of PHV in the introduction.
-Explain how authors can guarantee that the elderly who participated in this study made a sincere response. As mentioned in Section 2.4 in this study, an online video communication platform is used, and explain whether the use of the tool does not inhibit anonymity.
-Explain what AN and PP refer to in Chapter 2.
-Explain in more detail why authors adopted the development of question route used in reference [21].
-Explain how to solve 'obstacles to interpreting and communicating the information' in Chapters 3 or 4.
-In chapter 4, this study highlighted the importance of qualitative information over quantitative information. What efforts have been made in this study to collect high quality information?

Author Response

Dear Reviewer,

Sincerely, on behalf of the co-authors,

Anna Nivestam

Reviewer 2 Report

Thank you for the opportunity to review this work, which presents a cri de coeur to healthcare providers and policymakers to include the voices of older persons in practice and policy decision-making, using observations from key stakeholders and policymakers to tease out issues with and benefits of drawing on information from preventative home visits with older service users. There are a few minor revisions that could be made to augment the paper, though I would encourage both the editor and authors to pursue it to publication.

The authors have done an excellent job of establishing why this paper is of relevance and the approach it espouses is both worthwhile and necessary in the context of an aging world population and a more user-centered healthcare model. One thing that was not mentioned in the paper, but that would be of timely relevance, is a mention of the COVID-19 pandemic and its effects on older persons in the home, particularly in terms of a) service disruptions and b) government policies put in place on older people with little regard to their voices and personhood (e.g. Brooke, J., & Clark, M. (2020). Older people’s early experience of household isolation and social distancing during COVID‐19. Journal of clinical nursing, 29(21-22), 4387-4402, Armitage, R., & Nellums, L. B. (2020). COVID-19 and the consequences of isolating the elderly. The Lancet Public Health, 5(5), e256.) Adding a mention of this would highlight the timeliness and increase the relevance and contribution of the paper further. Although Preventative Home Visits (PHVs) is used in full in the abstract, it is not used in the introduction before commencing use of the shortened form (PHV). The full term should be used once (line 53) and then the acronym introduced for use from that point.

The authors have done an excellent job of not only explaining their method but justifying its selection and merits and provided clear descriptions of what they did and how. Concerning data analysis (line 210-220), it appears that a framework analysis method (Gale, N. K., Heath, G., Cameron, E., Rashid, S., & Redwood, S. (2013). Using the framework method for the analysis of qualitative data in multi-disciplinary health research. BMC medical research methodology, 13(1), 1-8) was perhaps used; could the authors clarify their content analysis approach?  

Additionally, it is appreciated that the authors included brief ethical considerations within their methods section. However, just because ethical approval is not legally required does not mean there are no ethical considerations, particularly in focus groups; while perhaps there are no anonymity issues here if participants are meeting in professional capacities and are known to each other, and fewer power considerations as the researchers saw each cohort (department heads/unit heads/politicians) separately, there are still issues of uneven and external dynamics and confidentiality inherent and potentially third-party harm; moderation of these focus groups would have to be done with an ethical approach of some sort in mind, and this should be addressed. (See for example Sim, J., & Waterfield, J. (2019). Focus group methodology: some ethical challenges. Quality & Quantity, 53(6), 3003-3022.)

The authors have conveyed findings clearly (the diagrammatic representation is a welcome addition). There is however something that does not quite sit right about the use of indirect, statistical information being considered a “voice” in the same way that a direct active qualitative voice might be; was this equivalence the view of the participants, or is it being established as equivalent by the researchers? If it is the latter, such equivalence would need to be justified with a reference. Otherwise, the Findings are relatively well presented, and make good use of the voice of all three different groups of participants to explain and expand on findings. One thing that stood out was that for each theme of the analysis only one set of discussion/quotes from one focus group was used; this made it unclear whether or not each focus group was asked about each theme, and whether or not each theme was applicable across the body of the data. Perhaps the authors might consider trimming down some of the quoted conversations to give more space to show other focus groups’ responses that complement the existing ones (e.g. 3.3.1 HoD’s are quoted about the better use of statistics; did politicians or HoU’s also talk about this?)

The discussion clearly draws our and questions or highlights key themes of the findings. In particular, the finding that more qualitative information is needed (around line 550) is really good, and potentially could be expanded slightly to really reinforce the value of this approach and point. The discussion is also missing any sort of discussion of whether or not a cultural context is at play here; are there particular Swedish cultural understandings of elder care that may make older persons' voices more or less likely to be heard in terms of their own care needs? Are there other cultural norms or considerations which get listened to first? (See, for example, Öberg, P., & Tornstam, L. (2003). Attitudes toward embodied old age among Swedes. The International Journal of Aging and Human Development, 56(2), 133-153.) This would be one of the unique offerings this paper could make, and form the basis of much future comparative work.

Author Response

(The authors gave the same response as above.)
